# Transition-Layer Implantation for Improving Magnetoelectric Response in Co-fired Laminated Composite

Sheng Liu [1] , Sihua Liao [1], Hongxiang Zou [1], Bo Qin [1,*] and Lianwen Deng [2]

1   College of Mechanical Engineering, Hunan Institute of Engineering, Xiangtan 411104, China
2   School of Physics and Electronics, Central South University, Changsha 410083, China
*   Correspondence: qinbo@hnie.edu.cn; Tel.: +86-13973255304

**Abstract:** Magnetoelectric (ME) laminated composites with strong ME coupling are becoming increasingly prevalent in the electron device field. In this paper, an enhancement of the ME coupling effect via transition-layer implantation for co-fired lead-free laminated composite ($80Bi_{0.5}Na_{0.5}TiO_3$-$20Bi_{0.5}K_{0.5}TiO_3$)/$(Ni_{0.8}Zn_{0.2})Fe_2O_4$ (BNKT/NZFO) was demonstrated. A transition layer composed of particulate ME composite 0.5BNKT-0.5NZFO was introduced between the BNKT piezoelectric layer and the NZFO magnetostrictive layer, effectively connecting the two-phase interface and strengthening interface stress transfer. In particular, an optimal ME voltage coefficients ($\alpha_{ME}$) of 144 mV/(cm·Oe) at 1 kHz and 1.05 V/(cm·Oe) at the resonant frequency in the composite was achieved, with a layer thickness ratio (BNKT:0.5BNKT-0.5NZFO:NZFO) of 3:1:6. The static elastic model was used to determine strong interface coupling. A large magnetodielectric (MD) response of 3.95% was found under a magnetic field excitation of 4 kOe. These results demonstrate that transition-layer implantation provides a new path to enhance the ME response in co-fired laminated composite, which can play an important role in developing magnetic field-tuned electronic devices.

**Keywords:** magnetoelectric coupling; laminated composite; transition layer; co-firing

## 1. Introduction

Multiferroics magnetoelectric (ME) material, offering inter-coupling among underlying magnetic and ferroelectric order parameters, has triggered increasing attention due to intelligent technological applications in the fields of detection, field control, implanted bioelectronics, storage devices, etc. [1–4]. ME coupling in most single-phase multiferroics is generally weak and only works at low temperatures, which renders it unsuitable for practical applications [4–7]. This shortcoming has motivated the development of synthetic, engineered composite multiferroics that have strong ME coupling at room temperature compared to their single-phase counterparts [2,8,9]. Depending on the connectivity of individual phases, ME composites are divided into three main categories based on their geometries: particulate (0-3 type), rod (1-3 type), and laminated (2-2 type) composites. Notably, laminate composites exhibit extremely high ME coupling responses. Laminated piezoelectric–magnetostrictive ME composites deliver great adjustability by controlling the composite configuration, microstructure, layer thickness or volume ratio, and interfacial structure [10,11]. Promising results have been experimentally and theoretically reported regarding ME interactions in laminated composites, such as PZT/Metglas [12], textured PMN-PT/$(Ni_{0.6}Cu_{0.2}Zn_{0.2})Fe_2O_4$ [13], Terfenol-D/PZT [14], PZT/$Mn_{0.4}Zn_{0.6}Fe_2O_4$ [15], and oriented PMN-PT/Ni plate [16]. Despite Pb-based ME systems presenting remarkable ME responses, they face inevitable limitations and are undesirable for applications due to the existence of lead toxicity and vaporization of PbO during sintering. Lead-free, eco-friendly, piezoelectric $80Bi_{0.5}Na_{0.5}TiO_3$-$20Bi_{0.5}K_{0.5}TiO_3$ (BNKT) shows high Curie temperature ($T_c$~320 °C), favorable piezoelectric response ($d_{33}$~195 pC/N), and low dielectric loss (tanδ < 0.05) [17,18], which is conducive to a large ME response.

In laminated ME composites, in addition to the intrinsic properties of the constituent piezoelectric and magnetostrictive phases, the interface quality significantly determines the strength of ME coupling. The interface bonding of laminated composite ceramics can be realized by co-firing or epoxy-bonding technology. Co-firing technology is widely used to synthesize epoxy-free, bonded, bulk laminated composites, such as $BaTiO_3/CoFe_2O_4$ [19], $(Ba_{0.85}Ca_{0.15})(Zr_{0.1}Ti_{0.9})O_3/NiFe_2O_4$ [20], $BiFeO_3$-$BaTiO_3/CoFe_2O_4$ [21], $Ba_{0.85}Ca_{0.15}Zr_{0.1}Ti_{0.9}O_3/La_{0.67}Ca_{0.33}MnO_3$ [22], and $BiScO_3$-$0.63PbTiO_3/NiFe_2O_4$ [23] ceramics. This technology involves the arrangement of oxide-based piezoelectric and magnetic layers in a suitable volume and the achievement of optimum density. However, the thermal shrinkage mismatch between magnetostrictive and piezoelectric constituent phases with different sintering thermodynamic behaviors in the co-firing process poses a challenge for the well-bonded interface, weakening strain-mediated coupling [24,25]. To relieve thermal shrinkage deficiency of the magnetostrictive and piezoelectric phases, liquid phase sintering became one of the main approaches. The introduction of sintering aids, such as $MnO_2$, $CuO$, and $Li_2CO_3$, effectively adjusted the thermal shrinkage of the oxide-based piezoelectric or magnetostrictive layer during co-firing and promoted the formation of a closely combined phase interface [19,26,27]. In spite of this, the interface microstructure in these systems is conducive to yielding large strain-mediated ME coupling due to the existence of excess liquid phase or its uncontrollable distribution at the phase interface. Moreover, the superfluous liquid phase causes an inhomogeneous grain gradient microstructure and impurity phase in the composite. Aiming to solve this problem, our group proposed an improved processing to establish a well-bonded interface for co-firing laminated ME ceramics by combining grain refinement and low-content liquid phase sintering [28,29]. The fine-grain and low-content sintering aid treatment produced a conforming sintering thermal shrinkage behavior for the piezoelectric and magnetostrictive layers at the interface and reduced thermal expansion mismatch between the two phases. It was also evidenced that the introduction of a suitable buffer layer between the composite layers provided the resulting well-bonded interface in the heterostructure and yielded effective strain transfer [30,31].

A method for selecting the insertion of a mixed transition layer with the same composition as the ferroelectric and magnetic phases in the laminated composite may be an efficacious way to bond the ferroelectric and magnetic phases and boost magneto-mechanical-electric coupling. The introduction of a mixed transition layer can effectively connect the two phases and alleviate interface delamination and microstructural defects caused by thermal shrinkage mismatch between the ferroelectric and magnetic phases. In this paper, ferroelectric BNKT and magnetostrictive $Ni_{0.8}Zn_{0.2}Fe_2O_4$ (NZFO) were specifically chosen to prepare laminated composites. A particulate ME composite (BNKT-NZFO) transition layer with a mass ratio of 1:1 served as the connection layer of the BNKT ferroelectric and NZFO magnetic phases. The microstructure, dielectric, ferroelectric, magnetic, as well as ME properties were studied. Further, a theoretical analysis was conducted to determine the correlation between ME voltage coefficient ($\alpha_{ME}$) and the degree of interface coupling.

## 2. Materials and Methods

The sol-gel method was employed for synthesizing the BNKT and NZFO powders. Analytical reagent grade $Bi(NO_3)_3 \cdot 5H_2O$, $NaNO_3$, $KNO_3$, $Ti(OC_4H_9)_4$, $Ni(NO_3)_2 \cdot 6H_2O$, $Zn(NO_3)_2 \cdot 6H_2O$, and $Fe(NO_3)_3 \cdot 9H_2O$ were used as raw materials. Deionized water was used as a solvent and citric acid was used as a complexing agent. Ammonia solution was used for neutralization. Details on the fabrication of NZFO and BNKT powders are described in ref. [32]. The sintered BNKT and NZFO powders were thoroughly ground together and ball-milled at a weight ratio of 1:1 (denoted as BNKT-NZFO) in ethanol for 16 h. To prepare the laminated ceramics, the individual ferroic powders and particulate composite powders were initially characterized to ensure their purity. The as-prepared BNKT powder was pressed inside a stainless steel die of 20 mm × 10 mm × 2 mm under a pressure of 60 MPa. Subsequently, the BNKT-NZFO composite powder was pressed on top of the BNKT layer as the transition layer under a pressure of 60 MPa. The as-prepared

NZFO powder was pressed on top of the transition layer under high pressure at 200 MPa. All the powders were mixed with 2% polyvinyl alcohol as a binder. According to the previous results, the maximal $\alpha_{ME}$ of the bilayer BNKT/NZFO composites appeared in the thickness ratio range of approximately 2:1 for the ferromagnetic and ferroelectric layers [27]. Therefore, the thickness $t_m$ of the NZFO layer, thickness $t_{mp}$ of the BNKT-NZFO composite layer, and thickness $t_p$ of the BNKT were 1.2, 0.2, and 0.6 mm, respectively. The layer thickness was adjusted by the amount of powder. Finally, the pressed green samples were sintered 2 h in air at 1100 °C. For the electric and ME measurements, the sintered pellets were painted with silver on the opposite side of the electrode and vertically polarized at an electric field of 5 kV/mm.

Phase identification was performed using Raman spectroscopy (BWS435) and X-ray diffraction (XRD) (Philips X-pert PRO). Grain structure and composition were analyzed using scanning electron microscopy (SEM) (Apreo S Lovac) and energy-dispersive X-ray fluorescence spectrometry (EDXRF). A Sawyer Tower circuit ferroelectric tester was used to measure the ferroelectric hysteresis loops and a vibrating sample magnetometer (Lakeshore 7404) was used to characterize the magnetic hysteresis loops. Magnetostriction measurement was characterized using a standard strain-gauge technique. The dielectric response and magnetodielectric value were measured using a precision impedance analyzer (Agilent 4294A) in a frequency range of 100 Hz to 1 MHz. The *dc* magnetic field source $H_{dc}$ was generated by an electromagnet. ME measurement was performed using a lock-in technique with a lock-in amplifier (SR830) by applying ac magnetic field ($H_{ac}$) superimposed with dc bias magnetic field ($H_{dc}$) [10,11]. The direction of the applied magnetic field was parallel to the surface of the sample and along the long side of the composite. A Helmholtz coil was driven by the functional signal generator to exert a small, continuous, sinusoidal, alternating magnetic field to drive a continuous ME response.

## 3. Results

The XRD patterns of the BNKT layer, BNKT-NZFO particulate composite, and NZFO layer from the co-fired composite are shown in Figure 1a. A set of typical diffraction peak characteristics clearly indicated the formation of a perovskite structure for BNKT and a cubic spinel structure for NZFO without any secondary phase. The XRD pattern of the BNKT-NZFO composite was disassembled into two groups of peaks, corresponding to the perovskite BNKT and spinel ferrite NZFO, respectively. No traces of additional or intermediate phase were identified. The lattice parameters of the cubic spinel NZFO in the ferrite layer and the BNKT-NZFO particulate composite were 8.399 and 8.401 A°, respectively. Splitting of the (200) and (002) peaks was observed in the BNKT phase, which was indicative of the presence of the morphotropic phase boundary [18]. Figure 1b depicts the Raman spectra of the NZFO layer, BNKT layer, and particulate composite for the cofired laminated sample. Raman active modes located at 132, 263, 299, 523, 671, 769, and 841 cm$^{-1}$ in the BNKT layer were in accordance with the modes of BaTiO$_3$-modified BNT [33,34]. Five active Raman modes situated at 301, 449, 542, 655, and 706 cm$^{-1}$ signified the formation of the cubic phase for NZFO. Further, the corresponding Raman modes of the individual phases were identified in the Raman spectra of the BNKT-NZFO particulate composite, confirming the presence of two separate phases remaining in the co-fired composite ceramic. This suggested that the introduction of transition-layer exerted no noticeable impact on the purity of the co-fired laminated ME composite. Figure 1c,d depict cross-section micrographs of the laminated sample. A dense microstructure with well-grown grains and well-bonded interface with no delamination at the BNKT/BNKT-NZFO and BNKT-NZFO/NZFO interfaces was observed. Thermal shrinkage of both ferroelectric and ferromagnetic components was simultaneously ameliorated by adding a transition layer. In addition, the element distribution in the laminated sample adjacent to the interface and away from the interface was recorded (Supporting Information Figure S1). Obviously, according to the distribution of Ti and Fe elements, the relative content of elements changed with the distribution of the ferroelectric and magnetic phase grains.

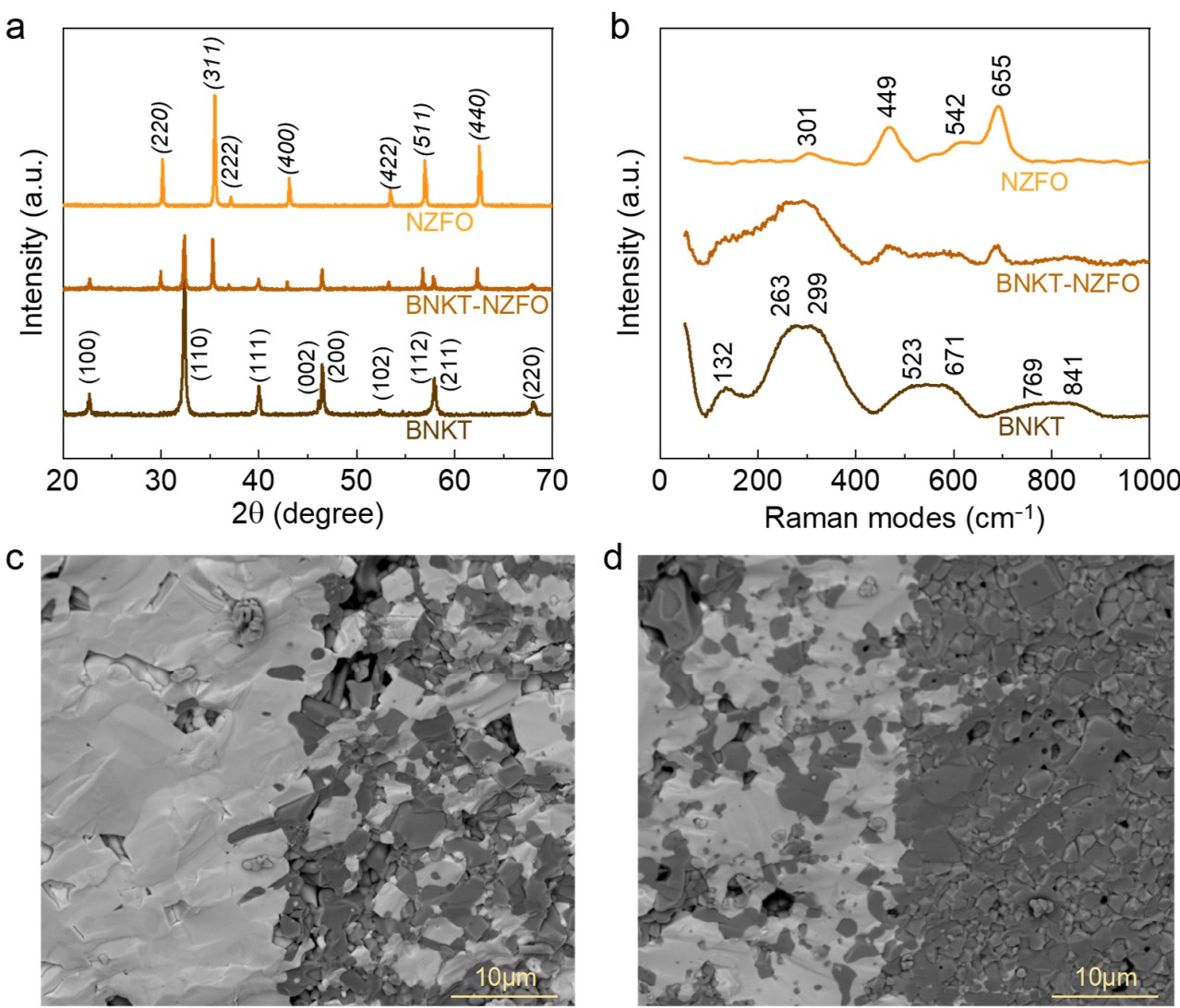

**Figure 1.** (**a**) XRD and (**b**) Raman spectra of NZFO layer, BNKT-NZFO composite, and BNKT layer. Cross-section morphology of the (**c**) BNKT/BNKT-NZFO interface and (**d**) BNKT-NZFO/NZFO interface.

To illustrate the ME response of the co-fired BNKT/BNKT-NZFO/NZFO composite, the dielectric, ferroelectric, and magnetic behaviors were separately investigated. Figure 2a plots the dielectric constant ($\varepsilon'$) and loss tangent (tanδ) of the BNKT, BNKT-NZFO composite, and NZFO at the frequency range 100 Hz to 1 MHz. The dielectric response of the composite exhibited a frequency dispersion, mainly attributed to the interface polarization occurring at the electrically heterogeneous interface. A significant electrical difference created Schottky barriers at phase interfaces, in which the inner field was formed by the charge concentration or depletion, causing large $\varepsilon'$ and tanδ at low frequencies. At high frequencies, the charge responses to the electric field were delayed, so a cut-off was formed in the frequency-dependent $\varepsilon'$ [35]. Figure 2b displays the polarization hysteresis ($P-E$) loops of the BNKT and laminated composite. The saturated hysteresis loops evidenced the presence of ferroelectric ordering. The observed remnant polarization ($P_r$) of 8.11 µC/cm$^2$ in the co-fired composite was found to be lower than the $P_r$ value (22.6 µC/cm$^2$) in a pure BNKT ferroelectric layer. The coercive field ($E_c$) of the co-fired sample was 22.9 kV/cm, which was larger than the value (18.6 kV/cm) observed for BNKT ceramics. The combination of BNKT with low-resistivity ferrite NZFO boosted the conductivity of the composite, making the electrical polarization insufficient and then degrading $P_r$. The increased $E_c$ was

attributable to the compounded non-piezoelectric layer, which electrically hardened the composite [36]. Figure 2c depicts the magnetic hysteresis ($M{-}H$) loops of ferrite NZFO and the laminated composite. All samples exhibited very compelling magnetic properties. The saturated magnetization $M_s$ was 34.94 emu/g for the laminated composite. The normalized $M_s$ (53.75 emu/g) in NZFO was comparable to the $M_s$ (52.52 emu/g) in the parent NZFO layer, signifying that the NZFO ferrite maintained its magnetic nature in the co-fired BNKT/BNKT-NZFO/NZFO composite. The dependence of the magnetostriction λ on $H_{dc}$ for NZFO was investigated, as shown in Figure 2d. The value of $λ_{11}$ reached −15 ppm at saturated $H_{dc}$ = 4000 Oe and the piezomagnetic coefficient $q_{11}$ (=d$λ_{11}$/d$H$) showed a maximum of $-16 \times 10^{-6}$ ppm/Oe at 720 Oe. This magnetostrictive deformation drove interfacial strain transfer and piezoelectric polarization.

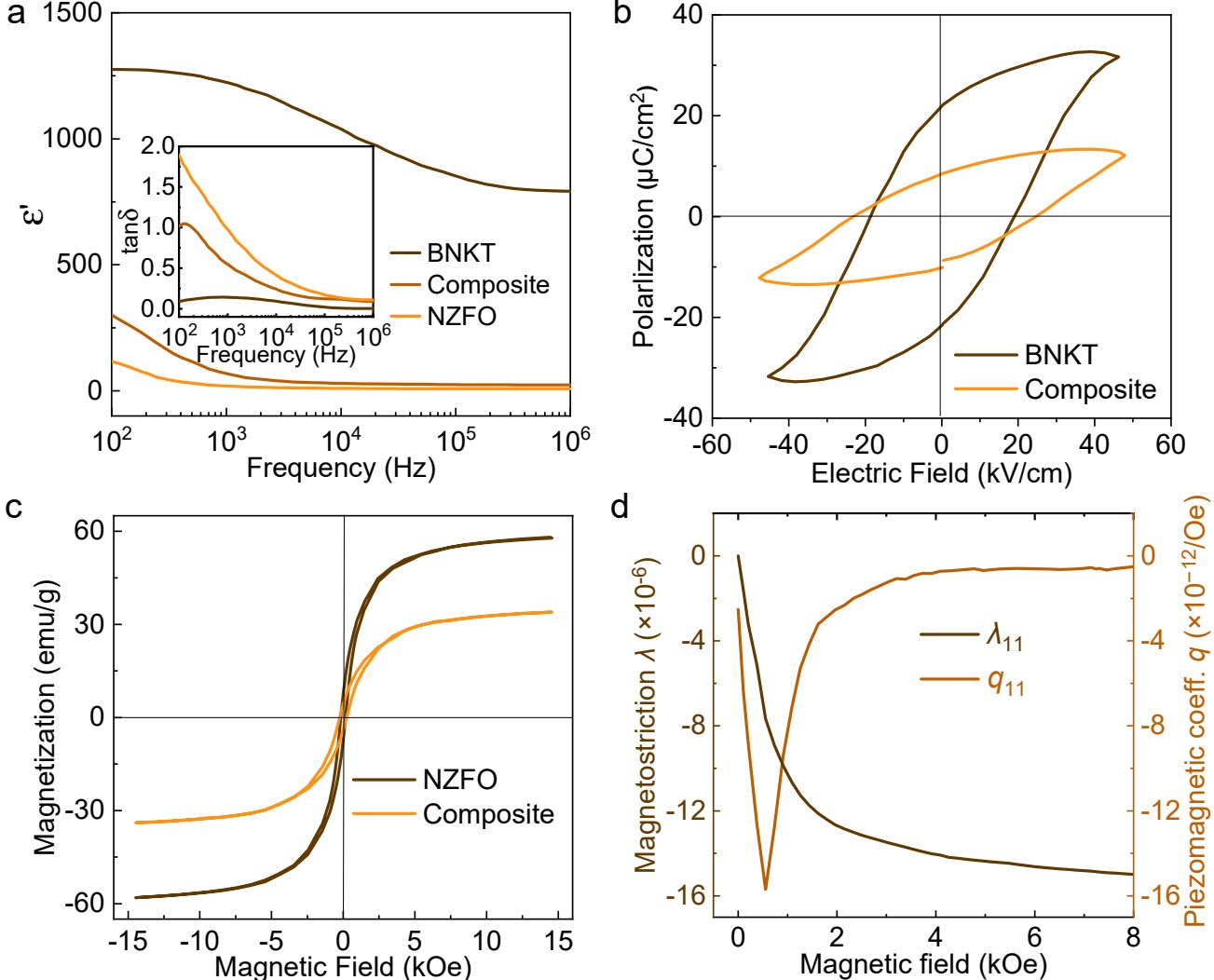

**Figure 2.** (**a**) Dielectric constant ($ε'$) and dielectric loss tangent (tanδ) and (**b**) ferroelectric polarization vs. electric field ($P{-}E$) hysteresis loops of the ferroelectric BNKT and composite. (**c**) Magnetic hysteresis loops of the ferrite NZFO and composite. (**d**) Variation in magnetostriction λ and piezomagnetic coefficient $q$ with *dc* magnetic field $H_{dc}$ for pure NZFO.

In light of the confirmation of the ferroelectric and magnetic properties, a measurable ME output could be envisaged in the BNKT/BNKT-NZFO/NZFO cofired composite. In an asymmetrical, two-layer, magnetostrictive-piezoelectric structure, bending and longitudinal deformations occur simultaneously with magnetic field excitation. The contribution from bending deformation is smaller than that from longitudinal deformation, especially for a

magnetic layer thickness much larger than that of the piezoelectric layer [37]. Therefore, in the BNKT/BNKT-NZFO/NZFO composite, we only considered longitudinal deformation experimentally and theoretically. Figure 3a shows the transverse $\alpha_{ME}$ curve (the magnetic field direction perpendicular to the polarization direction) as a function of magnetic field $H_{dc}$. We first clearly observed the relationship between the frequency and $\alpha_{ME}$, as shown in the inset of Figure 3b. A massive enhancement response occurred near 92 kHz for the BNKT/BNKT-NZFO/NZFO composite. Due to the electromechanical resonance effect, the interface strain transfer and its elastic coupling were significantly strengthened compared to the off-resonance frequency, resulting in a large ME response. It can be seen from Figure 3a that $\alpha_{ME}$ increased with an increase in $H_{dc}$, and then decreased for higher $H_{dc}$ after reaching a peak value near 750 Oe. When the external magnetic field gradually increased, the movement of the non-180° magnetic domain in the laminated composite was activated and the magnetic domain gradually entered a state of stress relaxation, resulting in the softening of the material and a decrease in the elastic modulus (negative $\Delta E$ effect). When the external magnetic field reached an optimum value, the negative $\Delta E$ effect was the most obvious and the $q$ of the composite was the largest. In this case, the stress transferred to the piezoelectric layer through the phase interface was the largest, leading to remarkable magnetic-mechanical-electric coupling. With the further increase in the external magnetic field, the binding effect of the external magnetic field on the magnetic domain was strengthened. At this time, the active state of the non-180° magnetic domain was suppressed and $q$ was reduced, which degraded the magnetic-mechanical-electric coupling. The non-monotonically variation tendency of $\alpha_{ME}$ was in line with the trend in $q$, revealing strain-mediated coupling in this composite. The maximal $\alpha_{ME}$ was observed with a large value of 144 mV/(cm·Oe) at 1 kHz and 1.05 V/(cm·Oe) at 1 kHz and the resonant frequency. This value was superior to that of co-fired, lead-free layered composites, such as bilayer $BaTiO_3/CoFe_2O_4$ (135 mV/cm·Oe@ 1 kHz) [19], $BiFeO_3$-$BaTiO_3/CoFe_2O_4$ (26.2 mV/cm·Oe@ 5 kHz) [21], $Ba_{0.85}Ca_{0.15}Zr_{0.1}Ti_{0.9}O_3/La_{0.67}Ca_{0.33}MnO_3$ (6.57 mV/cm·Oe@ 1 kHz) [22], $0.37BiScO_3$-$0.63PbTiO_3/NiFe_2O_4$ (60 mV/cm·Oe@ 1 kHz) [23], $Ba_{0.9}Ca_{0.1}Ti_{0.9}Zr_{0.1}O_3/Co_{0.8}Ni_{0.1}Zn_{0.1}Fe_2O_4$ (21.73 mV/cm·Oe@ 40 kHz) [38], $BaTiO_3$-$Bi_{0.5}Na_{0.5}TiO_3/BiY_2Fe_5O_{12}$ (6.6 mV/cm·Oe) [39], trilayer $CoFe_2O_4/(K_{0.5}Na_{0.5})_{0.96}Li_{0.04}Nb_{0.8}T_{0.2}O_3/CoFe_2O_4$ (8.3 mV/cm·Oe@ 10 kHz) [40], and multilayer $BiScO_3$–$PbTiO_3/NiFe_2O_4$ (108 mV/cm·Oe@ 10 kHz) [24].

As argued for the BNKT-based laminated composite [29], high-content ferrite produced a large compression deformation (negative magnetostriction for nickel ferrite) under applied $H_{dc}$ and transferred large strain to the piezoelectric BNKT through the interface, thus enhancing electric polarization. However, excessive ferrite thickness (or content) degraded the ferroelectric polarization, resulting in a relatively low ME output. Such synthetic action contributed an optimal value of $\alpha_{ME}$ at an optimized thickness ratio (ferrite layer: ferroelectric layer) of 2. Moreover, a zero-biased (self-biased) ME behavior was found in the $\alpha_{ME}$ curve. This zero-biased ME response was associated with the magnetization hysteresis of NZFO and the non-zero piezomagnetic coefficient in the low magnetic field range, which broadens the application field of BNKT/BNKT-NZFO/NZFO laminated composite for miniaturized electronic devices. To further confirm the ME interaction in the BNKT/BNKT-NZFO/NZFO laminated system, the variation of $\varepsilon'$ as a function of $H_{dc}$ is shown in Figure 3c. The $\varepsilon'$ increased with the increment of applied $H_{dc}$. Such a change in $\varepsilon'$ involved the magnetically controlled dielectric effect, involving an effective magnetodielectric (MD) effect. The MD effect is defined as $[\varepsilon'(H) - \varepsilon'(0)/\varepsilon'(0)] \times 100\%$, where $\varepsilon'(0)$ and $\varepsilon'(H)$ refer to the $\varepsilon'$ with and without the applied magnetic field, respectively. The MD value of the composite exhibited a continuous decrease with the increase in frequency and kept flat at high frequency. The observed MD behavior in the BNKT/BNKT-NZFO/NZFO composite was mainly associated with strain coupling between the BNKT and NZFO phases as a consequence of magnetostriction activity [27]. As shown in the inset of Figure 3b, the MD value showed an approximately quadratic increase with $H_{dc}$. This trend was reasonably scaled with the square of magnetization, indicating that the dielectric

susceptibility was associated with the square of the magnetic-order parameter. A large MD value of 3.95% (@ 1 kHz) was recorded at 4 kOe. This tunable MD behavior may provide potential applications in magnetic field sensing or imaging devices.

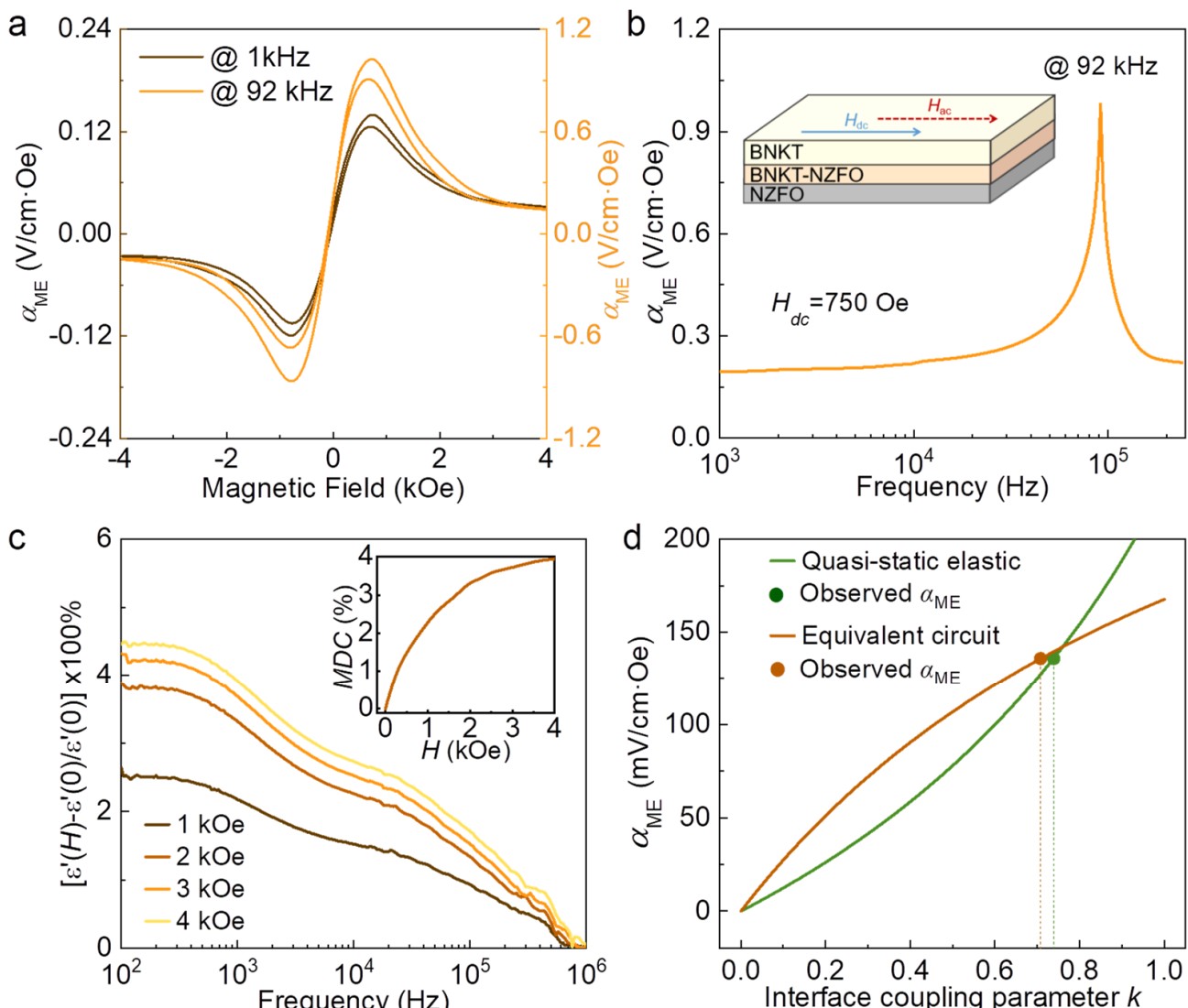

**Figure 3.** (**a**) Variation in ME voltage coefficient ($\alpha_{\mathrm{ME}}$) with *dc* magnetic field $H_{dc}$ for composites. (**b**) $\alpha_{\mathrm{ME}}$ dependent-frequency ranging from 1 kHz to 200 kHz at $H_{dc}$ = 750 Oe. Inset shows the co-fired laminated structure diagram. (**c**) Frequency-dependent magnetodielectric coefficient under the different *dc* magnetic field $H_{dc}$. Inset shows the dependence magnetodielectric coefficient on $H_{dc}$ ranging from 0 to 4 kOe at 1.0 kHz frequency. (**d**) Variation in the observed and calculated $\alpha_{\mathrm{ME}}$ values with interface coupling coefficient *k* for quasi-static elastic and equivalent circuit models at the frequency of 1 kHz.

To prove the interface coupling strengthening effect of the transition layer on the ME properties of the laminated composite, the interface coupling factor *k* was introduced to consider both the interface and ME effect in the non-ideal bilayer structure. The coupling factor *k* = 1 indicated that the piezoelectric and magnetostrictive phase interface were perfectly coupled (an ideal interface), while *k* = 0 indicated that no stress was transferred

at the interface. The $k$ value can be obtained from the quasi-static elastic and equivalent circuit models. The quasi-static elastic and equivalent circuit models are given as [41–43]:

$$\alpha_{ME} = \frac{-t_m t_p k \left({}^m q_{11} + {}^m q_{12}\right) {}^p d_{31}}{k {}^p \varepsilon_{33} \left({}^m S_{12} + {}^m S_{11}\right) t_p + {}^p \varepsilon_{33} \left({}^p S_{12} + {}^p S_{11}\right) t_m - 2k {}^p d_{31}{}^2 t_m} \tag{1}$$

$$\alpha_{ME} = \frac{t_m t_p {}^m q_{11} {}^p g_{31} k}{t_m {}^p S_{11} \left(1 - {}^p k_{31}{}^2\right) + t_p k {}^m S_{11}} \tag{2}$$

where superscripts $p$ and $m$ represent correlation parameters for piezoelectric and ferrite layers, respectively; $q_{ij}$, $d_{ki}$, $s_{ij}$, and $g_{ki}$ represent the effective piezomagnetic, piezoelectric coefficients, elastic compliance, and piezoelectric voltage constant, respectively; and $\varepsilon$ and $k_{ki}$ are the dielectric constant and electromechanical coupling coefficient, respectively.

The following parameters were used for the theoretical calculation: ${}^m s_{11} = 6.48 \times 10^{-12}$ m$^2$/N, ${}^m s_{12} = -2.37 \times 10^{-12}$ m$^2$/N, ${}^p s_{11} = 8.81 \times 10^{-12}$ m$^2$/N, ${}^p s_{12} = -4.95 \times 10^{-12}$ m$^2$/N, ${}^p d_{31} = -46.9$ pC/N, and ${}^p g_{31} = -43.5 \times 10^{-3}$ Vm/N [41,42,44]. $\varepsilon$, $q$, the thickness of piezoelectric layer $t_p$, and magnetostrictive layer $t_m$ were derived from the experimental value. ${}^p k_{31}$ was calculated via the equation ${}^p k_{31}{}^2 = {}^p d_{31}{}^2 / \varepsilon \varepsilon_0 {}^p s_{11}$. The calculated value of $\alpha_{ME}$ from the quasi-static elastic and equivalent circuit models as a function of $k$ was plotted for a particular thickness of $t_m = 1.3$ mm and $t_p = 0.6$ mm along with the observed values in Figure 3d. Notably, the calculated $\alpha_{ME}$ value was higher due to the assumption of 100% strain transfer efficiency at the interface ($k = 1$) and the approximation used in the calculation. The theoretically calculated maximum $\alpha_{ME}$ was estimated to be 221.4 mV/cm·Oe ($k = 1$ for quasi-static elastic model) and 171.2 mV/cm·Oe ($k = 1$ for equivalent circuit model), which were higher than those of the experimentally obtained maximum values. This discrepancy can be primarily attributed to deteriorated interface coupling as a consequence of microstructure defects from non-densification and slight inter-diffusion of the elements across the two layers during the high-temperature processing. According to the measured peak $\alpha_{ME}$ values of the BNKT/BNKT-NZFO/NZFO laminated composite, the $k$ values were 0.731 and 0.711, respectively. These obtained $k$ values were larger than those of epoxy-bonded bulk layered composite (0.11~0.13) and directly co-fired laminate composite ceramic (0.4~0.6) [28,29]. The transition-layer served as a connection layer to directly match piezoelectric oxide and magnetostrictive oxide layers, hence improving the interfacial strain quality. This result indicated that the transition-layer implantation route provided a stronger interface coupling effect than epoxying and direct co-firing, giving a pathway to enhance the ME effect in laminate composites. In this work, we only discussed the direct relationship between the interface coupling coefficient and $\alpha_{ME}$ under non-resonant conditions (@ 1 kHz). In future work, the influence of the resonant condition (bending resonance, longitudinal resonance, and their synergistic effect) on the interfacial coupling of laminated ME composites needs to be further studied experimentally and theoretically.

## 4. Conclusions

Laminated ME composite ceramics consisting of a lead-free ferroelectric BNKT layer, particulate ME transition layer (BNKT-NZFO), and a magnetostrictive NZFO layer were successfully prepared using a co-firing process. The introduction of a transition layer effectively connected the two-phase interface and strengthened interface coupling. A salient ME coupling of the laminated composite was observed with large $\alpha_{ME}$ ~144 mV/(cm·Oe) at 1 kHz and ~1.05 V/(cm·Oe) at the electromechanical resonance frequency. The improved interface coupling was demonstrated by the large interface coupling factor $k$. In addition, an MD response with a tunable value of ~3.95% was found in this laminated composite. These results indicate that the designed composite can be considered as an eco-friendly candidate for integration into magnetically controlled electronic devices, particularly transducers and magnetic sensors.

**Supplementary Materials:** The following supporting information can be downloaded at: https://www.mdpi.com/article/10.3390/magnetochemistry9020050/s1, Figure S1: Element distribution in laminated sample adjacent to the interface and away from the interface.

**Author Contributions:** Conceptualization, S.L. (Sheng Liu); Methodology, S.L. (Sheng Liu); Software, S.L. (Sihua Liao) and H.Z.; Validation, S.L. (Sheng Liu) and S.L. (Sihua Liao); Formal analysis, B.Q.; Investigation, S.L. (Sihua Liao) and B.Q.; Resources, S.L. (Sheng Liu); Data curation, B.Q.; Writing—original draft, S.L. (Sheng Liu); Writing—review & editing, H.Z., B.Q., and L.D.; Supervision, B.Q. and L.D.; Project administration, L.D.; Funding acquisition, H.Z. All authors have read and agreed to the published version of the manuscript.

**Funding:** This research was funded by the National Natural Science Foundation of China (grant no. 51902104) and the Scientific Research Fund of the Hunan Provincial Education Department (grant no. 22A0515).

**Institutional Review Board Statement:** Not applicable.

**Informed Consent Statement:** Not applicable.

**Data Availability Statement:** All data generated or analyzed during this study are included in this published article. The data used/analyzed are available from the corresponding author on request.

**Acknowledgments:** This research was supported by the National Natural Science Foundation of China (grant no. 51902104) and the Scientific Research Fund of the Hunan Provincial Education Department (grant no. 22A0515).

**Conflicts of Interest:** The authors declare no conflict of interest.

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
