# Peer review of "Transition-Layer Implantation for Improving Magnetoelectric Response in Co-fired Laminated Composite"

_magnetochemistry, doi:10.3390/magnetochemistry9020050_

Round 1

Reviewer 1 Report

The authors present results on magnetoelectric laminated composites using a cofired lead-free laminated structure. The result in the manuscript is interesting but some revisions are needed. Comments and suggestions given below.

Introduction: Add also a reference that discuss more material types (laminated structures) that have high magnetoelectric coupling at room temperature, such as the paper: Pedro Martins, Ana C. Lima, Victor A. L'vov, Nélson Pereira, Pimpet Sratong-on, Hideki Hosoda, Volodymyr Chernenko, Senentxu Lanceros-Mendez, In a search for effective giant magnetoelectric coupling: Magnetically induced elastic resonance in Ni-Mn-Ga/P(VDF-TrFE) composites, Applied Materials Today, Volume 29, 2022, 101682.

Page 2 row 45: add a sentence on the definition of the cofiring technique.

Page 2/3 section "Materials and methods": a suggestion is that the authors add a figure showing a cross section of the ferroelectric and magnetic phase with the thicknesses of each layer. This can be good for the reader.

Page 2/3 section "Materials and methods": How was the electrodes for the piezoelectric materials designed, placed and dimensions of the electrodes?

Page 3 row 108/109: Two comments, a) What coil system was used for the ac magnetic field?, b) It can be good to add a figure or some text explaining in what direction the dc and ac field was applied with respect to the sample.

Page 3 row 109: What LIA was used?

Page 4 row 124: Write "Five" instead of "5".

Page 4 row 150: For the final sample when investigating the ME effect, how were the samples poled (applied electric voltage, ramping the field, …?

Page 4 row 156 – 159: Two comments, a) From the hysteresis loops it looks like the sample exhibits hysteresis, with both remanence and coercivity. Add the values of remanence and coercivity in the text, b) Is it really saturated at 3000 Oe (when looking at figure 2d)?

Figure 2d: In the figure it looks like the data for lambda and q is switched. Check and correct this.

Page 6 row 194: Do the authors mean zero-biased by the magnetic remanence?

Equation 1: What does V3 stand for in equation 1 and 2 (is it the voltage)? The magnetoelectric coupling (alfa) is defined as the ratio of the electric field and applied magnetic field. In equation 1 and 2 it looks like the unit of alfa is V/Oe but it shall be V/(m*Oe). Check this and correct.

Page 6 and row 223: Two comments, a) do equation 2 takes into account mechanical resonance effects? b) What alfa value (at what frequency) is the calculated alfa value compared with?

Figure 3a: Two comments, a) Should not the unit on y-axis for figure 3a be V/(cm*Oe)? b) In figure 3a what means a negative ME coupling factor, is it a 180-degree phase shift between response and excitation?

Author Response

The authors present results on magnetoelectric laminated composites using a cofired lead-free laminated structure. The result in the manuscript is interesting but some revisions are needed. Comments and suggestions given below.

We thank you for carefully reading our manuscript and for giving detailed comments and suggestions that have been helpful to improve the manuscript quality.

Introduction: Add also a reference that discuss more material types (laminated structures) that have high magnetoelectric coupling at room temperature, such as the paper: Pedro Martins, Ana C. Lima, Victor A. L'vov, Nélson Pereira, Pimpet Sratong-on, Hideki Hosoda, Volodymyr Chernenko, Senentxu Lanceros-Mendez, In a search for effective giant magnetoelectric coupling: Magnetically induced elastic resonance in Ni-Mn-Ga/P(VDF-TrFE) composites, Applied Materials Today, Volume 29, 2022, 101682.

Response: Thank you for the suggestion. We have added relevant reference.

Martins, P., Lima, A.C., L'Vov, V.A., Pereira, N., Sratong-on, P., Hosoda, H., Chernenko, V. and Lanceros-Mendez, S. In a search for effective giant magnetoelectric coupling: Magnetically induced elastic resonance in Ni-Mn-Ga/P(VDF-TrFE) composites. Applied Materials Today 2022, 29, 101682.

Page 2 row 45: add a sentence on the definition of the cofiring technique.

Response: Thanks for your patient and detailed reviews. We have added a sentence on the definition of the cofiring technique.

Co-firing technology is the most suitable method for synthesizing epoxy-free bonded bulk laminated composites. This technology involves the arrangement of oxide-based piezoelectric and the magnetic layers in an appropriate volume and the achievementof  an optimum density.

Page 2/3 section "Materials and methods": a suggestion is that the authors add a figure showing a cross section of the ferroelectric and magnetic phase with the thicknesses of each layer. This can be good for the reader.

Response: Thank you for the valuable suggestion. We have added a schematic diagram to express the structure of laminated composites.

Page 2/3 section "Materials and methods": How was the electrodes for the piezoelectric materials designed, placed and dimensions of the electrodes?

Response: Thank you for your advice. The electrode treatment method is to cover the silver electrode on the surface of the piezoelectric material and the surface of the magnetic material.

Page 3 row 108/109: Two comments, a) What coil system was used for the ac magnetic field?, b) It can be good to add a figure or some text explaining in what direction the dc and ac field was applied with respect to the sample.

Response: We greatly appreciate the reviewer for comments and constructive suggestion. We have added text to describe the excitation magnetic field coil and its arrangement.

The ME measurement was performed using a lock-in technique with a lock-in amplifier (SR830) by applying ac magnetic field (Hac) superimposed with dc bias magnetic field (Hdc) and the induced voltage output was recorded. A Helmholtz coil was driven by the functional signal generator to exert a small and continuous sinusoidal alternating magnetic filed to drive a continuous ME response. The external dc magnetic field source Hdc was generated by an electromagnet.

Page 4 row 124: Write "Five" instead of "5".

Response: Thank you for the suggestion. We have replaced ' 5 ' with ' Five '.

Page 4 row 150: For the final sample when investigating the ME effect, how were the samples poled (applied electric voltage, ramping the field, …?

Response: For electric and ME measurements, the sintered pellets were painted with silver on the opposite side electrode and vertically polarized at an electric field of 5 kV/mm.

Page 4 row 156 – 159: Two comments, a) From the hysteresis loops it looks like the sample exhibits hysteresis, with both remanence and coercivity. Add the values of remanence and coercivity in the text, b) Is it really saturated at 3000 Oe (when looking at figure 2d)?

Response: Thanks for your patient and detailed reviews. We have added the values of remanence and coercivity in the manuscript.

The observed remnant polarization (Pr) of 8.11 μC/cm2 in the cofired composite is found to be lower than the Pr value (22.6 μC/cm2) in a pure BNKT ferroelectric layer. The coercive field (Ec) of the co-fired sample is 22.9 kV/cm, which is larger than the value (18.6 kV/cm) observed for BNKT ceramics.

When the DC magnetic field strength exceeds 3000 Oe, the values of magnetostrictive coefficient and piezomagnetic coefficient do not change significantly. Of course, the DC magnetic field strength greater than 4000Oe can basically reach saturation.

Figure 2d: In the figure it looks like the data for lambda and q is switched. Check and correct this.

Response: Thanks for your patient and detailed review. We have corrected the data caption for λ and q in Figure 2d.

Page 6 row 194: Do the authors mean zero-biased by the magnetic remanence?

Response: Thank you for carefully reading our manuscript and for giving detailed suggestions. We did not test the magnetoelectric coupling coefficient under zero bias field. In the next work, we will study the self-biased magnetoelectric coupling properties of layered magnetoelectric composites in detail.

Equation 1: What does V3 stand for in equation 1 and 2 (is it the voltage)? The magnetoelectric coupling (alfa) is defined as the ratio of the electric field and applied magnetic field. In equation 1 and 2 it looks like the unit of alfa is V/Oe but it shall be V/(m*Oe). Check this and correct.

Response: Thank you for the suggestion. We have checked the equation 1 and 2 and corrected the corresponding form of expression.

Page 6 and row 223: Two comments, a) do equation 2 takes into account mechanical resonance effects? b) What alfa value (at what frequency) is the calculated alfa value compared with?

Response: Thank you for the detailed comments. In the equation 2, we do not consider mechanical resonance effects. In the process of theoretical calculation, the calculated value of magnetoelectric voltage coefficient is obtained at the frequency of 1KHz. We only compare the theoretical and measured values of the magnetoelectric voltage coefficient at 1kHz.

Figure 3a: Two comments, a) Should not the unit on y-axis for figure 3a be V/(cm*Oe)? b) In figure 3a what means a negative ME coupling factor, is it a 180-degree phase shift between response and excitation?

Response: Thank you for carefully reading our manuscript and for giving detailed suggestions. In Figure 3a, the unit on y-axis should be V/(cm*Oe). We have changed the y-axis unit. In figure 3a, a negative ME coupling factor represent a 180-degree phase shift between response and excitation (reversely applied DC bias magnetic field).

Reviewer 2 Report

Comments and suggestions for the authors are in the attached file.

Author Response

We thank you for carefully reading our manuscript and for giving detailed comments and suggestions that have been helpful to improve the manuscript quality.

  1. Figure 2(d): The signatures to the curves lambda11 and q11 are mixed up.

Response: Thanks for your patient and detailed review. We have corrected the data caption for λ and q in Figure 2d.

  1. In Equations (1) and (2), it is not necessary to take the absolute value and there should be no signs of the differential “d”. It should be noted that the main difference between equations (1) and (2) is not that equation (2) is obtained using the modified equivalent circuit model, but that equation (1) takes into account deformations along the length and along the width of the ME sample, and equation (2) follows from a one-dimensional model that takes into account deformations are only along the length.

Response: Thank you for the suggestion. We have checked the equation 1 and 2 and corrected the corresponding form of expression.

  1. Line 220-222: the value pk31is not independent but is expressed by the
    equation pk312=pk312/εε0ps11. If you determine the value ε from experimental data, then you should not use the value pk31from the literature but should calculate it using the above equation.

Response: We greatly appreciate the reviewer for comment and constructive suggestion. We have removed the value pk31 from the literature and added the equation pk312=pk312/εε0ps11 for the parameter calculation.

  1. Figure 3(c, d): It is better to combine figures c and d into one figure. So it will be more convenient to compare. For a one-dimensional model, anequation similar to equation (2) can be obtained, only not for the quasistatic case, but for the resonant value of the ME voltage coefficient in the electro-mechanical resonance mode. It would be good to do this and make a figure similar Figure 3(d) only for the EMR mode.

Response: Thank you for the valuable suggestion. We have combined figures 3c and 3d into one figure.

Round 2

Reviewer 2 Report

Comments and suggestions for the authors are contained in the attached file.

Author Response

Comments and Suggestions for Authors

We thank you for carefully reading our manuscript and for giving detailed comments and suggestions that have been helpful to improve the manuscript quality.

1. Line 115-120: You still haven't answered another reviewer's questionabout how bias and alternating magnetic fields were directed in the  After all, it was easiest to give the appropriate figure here.

Response: Thank you for the suggestion. We have given the direction of bias and alternating magnetic fields in the experiment part. Also we have provided the appropriate figure in fig. 3(b).

2. Line 241: A typo in the equation. The equation should contain apiezoelectric coefficient in the right partpk312=pd312/εε0ps11.

Response: Thank you for the suggestion. We have corrected the equation pk312=pd312/εε0ps11.

3. Line 241-242: The meaning of the sentence is unclear. At what k, by whatequation, (1) or (2), is the theoretical calculatedME ? Why, if there is Figure 3(d)?

Response: Thanks for your patient and detailed reviews. We have provided the theoretical calculated ME value and the corresponding k value for quasi-static elastic model and equivalent circuit model.

4. You ignored my suggestion to make a figure similar to Figure 3(d) for theresonant mode, although the experimental data for the resonant mode aregiven in Figure 3(a,b). Explain why you don't want to follow my suggestion?

Response: Thank you for the valuable suggestion. In this work, we only experimentally measured the magnetoelectric coupling coefficient of layered magnetoelectric composites under resonance conditions, in order to confirm the enhancement of magnetoelectric coupling coefficient under non-resonant conditions and resonance conditions. The main idea of this paper is to prove that the introduction of transition layer improves the interface coupling and enhances the magnetoelectric coupling coefficient of layered composites. In the resonance model, we need to re-consider the interface coupling parameters and introduce the interface coupling parameters into the model. In the next work, we will use two models to calculate the magnetoelectric coupling coefficient under resonance conditions.

5. Apparently, the bias and alternating magnetic fields in the experiment weredirected along the long side of the composite. With such excitation of asymmetrical two-layer magnetostrictive-piezoelectric structure, the longitudinal and bending modes of the magnetoelectric effect are simultaneously excited. In the resonance mode, only the longitudinal mode can be considered separately. But in the quasi-static mode, it is necessary to take into account the contribution from both the longitudinal and bending  Please look at the article https://www.mdpi.com/2504-477X/5/11/287 for an example. Equations (1) and (2) take into account only the contribution of the longitudinal mode. To obtain the correct result, the contribution of the bending mode must be taken into account. Moreover, in the context of your research, it should take into account the imperfection of the mechanical connection of the two phases and contain the appropriate parameter k.

Response: Thank you for carefully reading our manuscript and for giving detailed suggestions. The article “https://www.mdpi.com/2504-477X/5/11/287” has provided a model for the low-frequency magnetoelectric effect that takes into consideration the bending and longitudinal deformations in a ferromagnetic and ferroelectric bilayer. It does give us a lot of inspiration. Bending resonance affects the magnetoelectric behavior of layered magnetoelectric composites. Our work focuses on the introduction of transition layer to improve the interface coupling of layered magnetoelectric composites. The influence of bending resonance, longitudinal resonance and their synergistic effect on the interfacial coupling of layered magnetoelectric composites needs to be further defined experimentally and theoretically. We will focus on the preparation of layered magnetoelectric composites in the next step. Using the model proposed in the article, the interface coupling coefficient will be introduced to extract a more accurate magnetoelectric coupling model considering the degree of interface coupling. In our research, we have taken into account the imperfection of the mechanical connection of the two phases and described the discrepancy.